# Multi-Copy Relay Node Selection Strategy Based on Reinforcement Learning

**DOI:** 10.3390/s23136131

**Published:** 2023-07-04

**Authors:** Yang Gao, Fuquan Zhang

**Affiliations:** College of Information Science and Technology, Nanjing Forestry University, Nanjing 210000, China; gaoyang@njfu.edu.cn

**Keywords:** Q-lambda algorithm, community division, relay node, structural similarity, node degree, interests

## Abstract

Delay tolerant networks (DTNs), are characterized by their difficulty in establishing end-to-end paths and and large message propagation delays. To control network overhead costs, reduce message delays, and improve delivery rates in DTNs, it is essential to not only delete messages that have reached their destination but also to more precisely determine appropriate relay nodes. Based on the above goals, this paper constructs a multi-copy relay node selection router algorithm based on Q-lambda reinforcement learning with reference to the idea of community division (QLCR). In community division, if a node has the highestdegree, it is considered the core node, and nodes with similar interests and structural properties are divided into a community. Node degree refers to the number of nodes associated with the node, indicating its importance in the network. Structural similarity determines the distance between nodes. The selection of relay nodes considers node degree, interests, and structural similarity. The Q-lambda reinforcement learning algorithm enables each node to learn from the entire network, setting corresponding reward values based on encountered nodes meeting the specified conditions. Through iterative processes, the node with the most cumulative reward value is chosen as the final relay node. Experimental results demonstrate that the proposed algorithm achieves a high delivery rate while maintaining low network overhead and delay.

## 1. Introduction

Since the concept of the DTN (Delay Tolerant Network) was introduced, its application fields have continuously expanded with the deepening of research; these include traffic networks [1], remote area networks [2,3,4], earthquake emergency rescue networks [5], battlefield networks [6], etc. These networks can facilitate data transmission in harsh communication environments.

The main research areas in DTNs include DTN architecture [7], routing design [8], congestion control [9], security mechanisms [10], mobility models [11], and simulation platforms [12]. Despite significant changes in the connotation and denotation of DTNs, their core essence remains unchanged: all nodes in the network utilize opportunistic encounters for message transmission. Routing design is a hot research topic, as a well-performing routing algorithm can deliver messages to the destination node in a short time with minimal overhead. Designing an efficient routing algorithm is a key and challenging problem in DTN [13].

In Delay tolerant networks, no reliable links exist between source and destination nodes, and message transmission relies on node mobility [14]. Consequently, routing protocols play a crucial role in network performance. DTN architecture differs significantly from TCP/IP, forming a network structure with “store-carry-forward” characteristics [15]. However, node size and weight are generally small, limiting their storage space and communication capabilities, and thus constraining the energy efficiency of applications.

Moreover, network topology changes continuously due to node mobility, resulting in varying numbers of nodes encountered within communication ranges. Flooding or message copying for message transmission consumes significant network resources, potentially causing network congestion. One solution to overcome this limitation is to more precisely identify corresponding relay nodes within encountered nodes and deliver messages in a timely manner.

Nodes in DTNs typically refer to people and various mobile devices, with most mobile devices carried by people, making the network predominantly human-based. Node mobility enables interaction among people, forming social relations. In contrast to the rapid changes in node movement speed and network topology, social relationships between nodes are relatively stable, interdependent, and rule-based [16]. Stable social relationships form social networks, where people with similar social behaviors create various community structures. Individuals within the same community structure are closely connected, often sharing similar interests, hobbies, and behavioral characteristics [17].

DTNs are characterized by unreliable and unstable connections, where nodes may experience long periods of unreachability, high latency, or unpredictable link disruptions. This scenario is similar to a wireless sensor network [18]. The selection of relay nodes can help improve network connectivity and ensure that messages can be delivered through multi-hop paths, thereby increasing the success rate of message delivery. In DTNs, due to the instability of network topology and the unreliability of information, it is not possible to accurately obtain complete network topology and real-time routing information. Therefore, relay node selection can make routing decisions based on the current network conditions and local information, enabling effective routing selection. Nodes in DTNs typically have limited resources such as storage space, bandwidth, and energy. By selecting appropriate relay nodes, network resources can be efficiently utilized, avoiding resource wastage and energy consumption, thereby improving overall network performance and efficiency. Message delivery in DTNs often requires multiple relays to reach the destination, so the selection of relay nodes directly affects the efficiency and delay of message delivery. Choosing suitable relay nodes can shorten the message delivery path and reduce transmission delays, thereby enhancing message delivery efficiency. In conclusion, relay node selection is of significant importance in DTNs as it improves network connectivity, enables effective routing selection, optimizes resource utilization and efficiency, and enhances message delivery efficiency. These factors are crucial for achieving reliable communication in unreliable and high-latency environments.

In this paper, we propose a DTN routing algorithm based on Q-Lambda reinforcement learning inspired by community division. The Q-Lambda algorithm is used to determine the most suitable relay node for message delivery. Additionally, when nodes encounter each other, they promptly delete successfully delivered messages to save cache space. Experimental results indicate that our method outperforms traditional approaches.

## 2. Related Work

The purpose of employing the concept of community division to identify the optimal relay node is to facilitate timely message transfer, thereby reducing network overhead and congestion probability [19]. This approach increases the possibility of successful delivery, achieving a higher message delivery rate than traditional routing algorithms. The Epidemic Router (ER) algorithm utilizes a flooding method for information transmission, maximizing the success rate of information transmission and minimizing transmission delay [20]. However, the excessive number of messages in the network can consume substantial network resources [21]. To reduce the number of messages in the network, researchers have proposed routing algorithms that limit the number of message copies, such as Spray And Wait (SAW) and Prophet Router (PR) algorithms. These classic DTN routing algorithms each have their own strengths and can be selected according to specific needs in various environments. In order to avoid blindly forwarding messages to relay nodes, a hybrid of message delivery probability algorithm (HPR) is proposed in paper [22], which adopts the transmission probability value of nodes as the basis for forwarding messages. Consequently, messages are consistently forwarded in the direction of increasing delivery probability values, helping to reduce overhead.

Since the social relationships between nodes are stable, interdependent, and follow certain rules, we propose a method to determine whether a node is suitable as a relay node according to the degree of interest of each node in different regions, the structural similarity of nodes, and the degree of nodes. Combined with the Q-Lambda reinforcement learning algorithm, the optimal relay node selection problem is addressed.

Regardless of whether it is a traditional community division algorithm or a newly proposed one, the ultimate purpose of community division is to group nodes with the same interests and hobbies, as well as those with close connections and similar behavioral characteristics, into the same community [23,24,25,26,27,28]. The community division algorithm proposed by Niu Dongdong et al. designates the high node degree as the core node of a community and then divides the remaining nodes into corresponding communities through a similarity algorithm [29]. The greater the structural similarity of nodes, the higher the probability they are in the same community, further proving that the relationship between them is stronger. Community division research can help us better understand complex network relationships.

Since each node maintains a queue with a level of intimacy, it consumes more resources and increases network overhead. To address this issue, we promptly delete messages that have reached their destination nodes.

In recent years, DTN routing improvement algorithms have emerged one after another; with the continuous development of machine learning technology, more possibilities have been brought to the routing algorithm design of delay tolerant networks [30,31,32,33,34]. In 2013, Rolla et al. proposed using a multi-agent reinforcement learning algorithm to solve the message forwarding problem, employing a multi-agent Q-learning algorithm to forward message copy knowledge to generate the optimal reward node calculated by the distance function between nodes [35]. Its reward value is related to the distance between nodes. Utilizing the maximum expected reward value as the best next hop, message forwarding strategy selection and Q table update are conducted simultaneously. However, this approach can easily overestimate the reward value of nodes. Q-learning is intended to converge to an optimal policy by iteratively updating Q values based on a given reward signal. However, convergence is not guaranteed, and the learning process can be sensitive to initial conditions and parameter settings. In some cases, Q-learning may become stuck in suboptimal solutions or exhibit unstable learning behavior. Therefore, this paper introduces the Q-Lambda algorithm, a classical Q-learning method based on discrete Markov decision processes, combined with a homeostatic division method for multi-step returns and trace extraction algorithm, to address the shortcomings of reinforcement learning algorithms with poor local ability to find the best next hop. We have provided a brief introduction to the algorithms mentioned above in Table 1.

This paper proposes a DTN routing method based on Q-Lambda reinforcement learning, which refers to the concept of community division to determine appropriate relay nodes. Our contributions are as follows:We designed a Q-Lambda-based reinforcement learning router algorithm, which references the idea of community partitioning to determine appropriate relay nodes (QLCR).We planned the movement route of nodes and set corresponding points of interest according to the actual situation using node degree, interest, and the structural similarity combination of decision.We have carried out simulation experiments on the algorithm, and the experimental results show that this algorithm is superior to other algorithms.

## 3. Proposed Method

### 3.1. System Model

In this article, we simulate a scenario where different types of people move within various ranges. The communication range of each node is 5 m, as shown in Figure 1. Only when two nodes are within each other’s communication range will they establish a connection. The number of nodes in the map is m, and the set of nodes is *M*. The information exchange between nodes is expressed as $N2NC_{i}=\left\{1,2,\ldots ,m\right\},i\in M$. Each N2NC pair has a transmitter and a receiver implementation to send and receive messages. Through this system model, we can simulate message propagation in delay tolerant networks. The proposed strategies will be applied in this model to achieve more efficient information delivery.

In this paper, the entire experimental area is divided into five different areas of interest (t0–t4) according to their specific functions. The population is divided into different types (students, residents, teachers, others) and the movement range of different types of people was limited. Each node records the number of times it reaches each area, and the recorded times represent the degree of interest in a certain area. When nodes carrying messages forward messages, those most interested in the same region are selected for forwarding messages [36]. Some symbols are defined as follows: G = (M, E), where M represents the set of points and E represents the set of edges; M(v) denotes the set of neighbors of node v and $d_{v}$ represents the degree of node *v*. Node degree refers to the number of edges associated with a node, represented by d. The larger the value of d, the more nodes are connected to the current node, and the larger the degree of a node, the more important the node is within the network [37]. There is a high probability that such a node is in a key position and the message is forwarded frequently. Therefore, the importance of a node is generally measured by the size of its degree. Forwarding a message to a node with a high degree increases the probability that the message will reach its destination. In this paper, the size of the node degree is used as one of the conditions to determine the relay node.

Ideas were divided by community; the nodes within a community are much denser than those between communities, which implies that each node and the majority of its neighbors should belong to the same community. Consequently, if we employ the neighborhood correlation metric to calculate the similarity sim(m,n) between any pair of connected nodes (m,n), the similarity between nodes within the same community will typically be much greater than the similarity between nodes situated at the boundaries of different communities. With this in mind, the similarity calculation formula is as follows [38].
(1)$$\begin{array}{c} Sim\left(m,n\right)=\frac{\left|N\left(m\right)\cap N\left(n\right)\right|}{d_{m}} \end{array}$$

It is close to the similarity measure proposed in [39], but this similarity measure is symmetric. However, in many practical applications, asymmetric similarity measures can better describe node relationships. For example, m has a large social circle, while *n* has very few friends. If *m* and *n* are friends, due to *n* having fewer friends, *n* will consider *m* very important. On the contrary, due to *m* having many friends, *m* may consider *n* less important. So we have the following formula:(2)$$\begin{array}{c} Sim\left(n,m\right)=\frac{\left|N\left(n\right)\cap N\left(m\right)\right|}{d_{n}} \end{array}$$

Based on this phenomenon, asymmetric similarity is defined as sim(m,n)!=sim(n,m). Structural similarity is also used as a criterion for selecting a suitable relay node.

The Q-Lambda algorithm is improved on the basis of a reinforcement learning algorithm, introducing memory function. This algorithm iterates over the recorded state actions repeatedly. We want the system to calculate and return a reward value *R*. Therefore, it is necessary to create a *Q* table to store *Q* values. Its iterative formula is as follows [40]:(3)$$\begin{array}{c} Q_{t+1}\left(s,a\right)\leftarrow \;lambda\times \left[R+\gamma \times max_{s\in A}Q_{t+1}\left(s_{i},a\right)\right]\\ +\left(1-\;lambda\right)\times Q_{t}\left(s,a\right) \end{array}$$

In the above formula, γ is the reward attenuation factor and lambda is the learning efficiency. *A* represents the set of all points encountered by node a. It is not difficult to see that $s_{i}$ is the node that has met node *a*. $max_{s\in A}Q_{t+1}\left(s_{i},a\right)$ represents the maximum value in the *Q* table of node a. Usually lambda and gamma are fixed. The reward value *R* is affected by node degree, structural similarity, and region of interest.

We can consider the *Q* function as a reader that scans the *Q*-table, searching for rows associated with the current message state and columns associated with the message action. It returns the *Q* value from the corresponding cell, which represents the expected future reward (*Q*-value). The *Q*(λ) algorithm is employed to determine the most suitable relay node for forwarding messages. The learning environment for the *Q*(λ) algorithm encompasses the entire network, with each node serving as a learning agent that can acquire knowledge from the current network by delivering messages to the nodes it encounters.

### 3.2. Calculation of Reward Value

On a DTN, a large number of messages are copied, causing resource shortage and network congestion. So, we adopt the method of excellent relay node selection to improve the overall performance of the network. The calculation method for structural similarity has been introduced in Equations (1) and (2); Degree (Degree A represents the degree of current node A; Degree B represents the degree of potential relay node B); and Region Matching (Region A represents the region of greatest interest to current node A; Region B represents the region of greatest interest to potential relay node B). In the algorithm, the reward value is set to the following three conditions:

If (Structural Similarity(A, B) > Threshold and Degree(B) > Degree(A) and RegionMatch(A, B))

R = 3

If (Structural Similarity(A, B) < Threshold and Degree(B) > Degree(A) and RegionMatch(A, B))

R = 2

If (Structural Similarity(A, B) < Threshold and Degree(B) < Degree(A) and RegionMatch(A, B))

R = 1

### 3.3. Update the Q Value in the Q Table

The Q table updates are presented in Algorithm 1. When two nodes enter the communication range, a connection is established between them, resulting in the acquisition of the initial state–action pair. Upon subsequent encounters between nodes, an iterative operation is executed according to Equations (1) and (2), which includes calculating the structural similarity between the two nodes, comparing the node degrees, and determining whether their regions of interest are the same.
**Algorithm 1** Update the Q table  1: ▹R:rewardvalue  2: ▹Lambda:Timeattenuationparameter  3: ▹Q(s,a):StoragenodeQvalue  4: ▹gamma:Attenuationcoefficient  5: $▹\;d:\;\mathrm{Node}\;\mathrm{degree}.\;a_{d}\;\mathrm{represents}\;\mathrm{the}\;\mathrm{degree}\;\mathrm{of}\;\mathrm{node}\;a,\;s_{d}\;\mathrm{represents}\;\mathrm{the}\;\mathrm{degree}\;\mathrm{of}\;\mathrm{node}\;s$  6: ▹S:NodeStructuralsimilarity  7: $▹\;i:\;\mathrm{Region}\;\mathrm{of}\;\mathrm{interest}.\;a_{i}\;\mathrm{represents}\;\mathrm{the}\;\mathrm{region}\;\mathrm{of}\;\mathrm{node}\;a,\;s_{i}\;\mathrm{represents}\;\mathrm{the}\;\mathrm{region}\;\mathrm{of}\;\mathrm{node}\;s\;$  8:**while** $\left(connection_{sa}.isUp\right)$**do**  9:     R←010:     **if** ($a_{d}>s_{d}\wedge S<threshold\wedge a_{i}==s_{i}$) **then**11:          R←312:     **end if**13:     **if** ($a_{d}>s_{d}\wedge S<threshold\wedge a_{i}==s_{i}$) **then**14:          R←215:     **end if**16:     **if** ($a_{d}<s_{d}\wedge S<threshold\wedge a_{i}==s_{i}$) **then**17:          R←118:     **end if**19:     Update the Q value in the Q table;20:     **if** (The update Q value is the Maximum value in the Q table) **then**21:         Messages transfer;22:     **else**23:         Do not transfer messages;24:     **end if**25:**end while**

Research on community structure can uncover valuable information and alerts researchers to the accuracy and reliability of the methods employed to detect these substructures. A significant development in community detection originates from [41], where a quality measure [42], modularity (G), is proposed by the same author to quantify modular structures and assess the merits of each community division. When the value after the partition is larger than the value before the partition, the community partitioning is deemed satisfactory. Hence, we posit that when two nodes meet and the calculated Q value is larger than the maximum reward value in the current Q table, the message will be forwarded.

Figure 2 shows the updating process of Q values in the Q table. In Figure 2a, node A and node B, respectively, calculate Q values after establishing connections and fill them into the Q table maintained by them. Since there are no other nodes in the table, the maximum value in table Q is the current value, and then they will pass messages to each other. In Figure 2b when nodes A and B meet again, they calculate Q again, overwriting the previous value. However, the value of B in node A is less than the maximum reward value in the whole Q table. So, A does not pass messages to B. In node B, node A’s Q value is the largest, so then node B will pass the message to A.

### 3.4. Node Buffer Management

DTN routing policies can be divided into single-copy and multi-copy policies [43]. While the former consumes fewer resources, it struggles to achieve successful message transmission. To enhance message delivery rates and reduce transmission delays, multi-copy policies are typically adopted. This algorithm adopts a first in, first out message transmission mechanism within a link time. Under a multi-copy strategy, due to factors such as price and volume, the limited cache space of nodes cannot accommodate the storage requirements for numerous message copies. Consequently, some message copies are discarded during transmission and fail to reach the destination node. This leads to a decrease in the overall network delivery rate. Therefore, devising effective cache management policies, classifying and evaluating different message copies, and storing and deleting message copies in a targeted manner are crucial to improving the overall performance of the network.

In a DTN, each node maintains a record of successful message deliveries. Thus, when two nodes encounter one another, they can delete the messages that have been successfully delivered. This approach saves storage space, improves message delivery rates, and reduces network overhead. The buffer management algorithm is demonstrated in Algorithm 2.
**Algorithm 2** Manage node cache  1:**if** (this node’s acklist is not empty) **then**  2:      Encounter node UpdateList();  3:      Locate and delete messages;  4:**end if**  5:**if** (encounter node’s acklist is not empty) **then**  6:      UpdateList();  7:      Locate and delete messages;  8:**end if**

## 4. Results

The simulation of the proposed algorithm was achieved on the basic simulation platform of the Opportunistic Networking Environment (ONE) in this section. We use the ONE simulation platform to evaluate several classical algorithms and proposed algorithms and show the results of their comparison.

As the range of campus is smaller than that of the map provided by ONE, the number of nodes ranges from 30 to 180.The interval for message generation is from 5 s to 35 s.Speed options range from 5 m per second to 20 m per second.The cache is 5 M to 35 M.

We recorded the basic performance indicators (delivery rate, message delay, network overhead, average hops) in the DTN, and the simulation parameters were configured as shown in Table 2.

The overhead ratio is used to evaluate the effectiveness of message replication in DTNs. Redundant forwarding of message copies can waste network resources and cause network congestion. The lower the network overhead, the fewer redundant relay transmissions occur in the DTN, indicating better network performance.

The message delay is used to evaluate the transmission speed of DTNs. The unit of message delay is second. The message delay measures the time required for a message to travel from the source node to the destination node. A lower message delay indicates faster message transmission in the network, while a higher message delay may suggest issues such as network congestion or unstable paths. Therefore, message delay is one of the important metrics for assessing the performance and efficiency of DTNs.

In a DTN, the comparison results of five different routing protocols (QLCR, PR, ER, HPR, and SAW) are shown in Figure 3, Figure 4, Figure 5 and Figure 6.

### 4.1. Analysis of Message Generation Interval

As shown in Figure 3a, the delivery rate of all messages increases as the message interval grows. This is because the number of messages in the network decreases as the interval between message generation increases. When the interval is 35, the delivery rate of the proposed algorithm is 0.9052, which is 8.8% higher than that of the second-ranked SAW algorithm (0.8250), and 81.8% higher than that of the lowest-performing ER algorithm.

When TTL = 5, the proposed algorithm has a 54% delivery rate, which is also the highest delivery rate among all algorithms. As can be seen from Figure 3b, the cost of the proposed algorithm decreases gradually, primarily because messages are deleted in time to save space. Except for the SAW algorithm, the overhead of other algorithms increases gradually. As shown in Figure 3c, the message delay of the proposed algorithm is minimal. The message delay of these algorithms increases as the TTL value grows.

### 4.2. Node Movement Speed Analysis

As shown in Figure 4a–c, the delivery rate first rises and then falls as the speed increases, indicating that our proposed algorithm is best suited for medium–low-speed moving networks.

When the speed is too high, nodes encounter each other more frequently, which leads to a large number of duplicate messages in the network. As a result, network congestion is more likely to occur. This is evident in the decline of delivery rates for the other algorithms when the speed is 20. Rapid node movement can also cause unstable connections between nodes and potentially reduce the delivery rate.

### 4.3. Node Number Analysis

As demonstrated in Figure 5a, the proposed algorithm consistently achieves better delivery rates under any number of nodes. Apart from the SAW algorithm, the delivery rates of the other three algorithms decrease as the number of nodes increases.

From Figure 5b, we can see that the number of nodes has minimal impact on the overhead. Figure 5c shows that as the node count increases, the message delay of the proposed algorithm gradually decreases, indicating that the proposed algorithm can adapt to complex networks.

### 4.4. Node Cache Analysis

Typically, as the node cache size increases, so does the number of messages stored by a node. Figure 6a–c reveals that the performance of PR, ER, and HPR algorithms improves with larger caches. The proposed QLCR algorithm and SAW algorithm have limitations on the number of message copies, making them adaptable to a wider range of networks.

## 5. Discussion

From the experimental results, it can be seen that using the Q-lambda reinforcement learning algorithm demonstrates certain advantages over other algorithms in certain scenarios. Furthermore, our proposed algorithm achieves higher message delivery rates with lower latency. This indicates that the Q-lambda algorithm has a certain advantage in handling tasks with latency requirements. Exploration is a key aspect in reinforcement learning for discovering potential rewards and optimal strategies. The Q-lambda algorithm maintains exploratory behavior to better discover and exploit potential rewards in the environment. This exploratory characteristic aids in discovering new and superior strategies, thereby enhancing the delivery rate.

Our proposed QLCR algorithm implements an efficient routing selection method for network communication. Compared to the SAW algorithm, QLCR outperforms it in achieving maximum delivery rate while enabling data transmission with the lowest latency. In Figure 3d, Figure 4d, Figure 5d and Figure 6d despite not having the lowest average hop count, our algorithm remains within an acceptable range, ensuring efficient and reliable network communication.

It is important to note that the effectiveness of an algorithm largely depends on the specific application scenario and task requirements. Although QLCR has shown superior performance over other algorithms in some experiments, it does not imply that it is the optimal choice in all situations. In practical applications, it is necessary to consider the algorithm’s performance, efficiency, and suitability, and select the most appropriate reinforcement learning algorithm for the specific task.

## 6. Conclusions

This paper proposes a DTN routing method based on Q-Lambda reinforcement learning and incorporates the concept of community division to identify suitable relay nodes. To enhance the overall performance of the network, we implement timely deletion and delivery of successfully transmitted messages. We compare the QLCR algorithm with traditional routing algorithms and the HPR algorithm. From Figure 3, Figure 4, Figure 5 and Figure 6, it is evident that the proposed algorithm exhibits significant improvements in message delivery rate, overhead, delay, and other performance metrics. Notably, the QLCR algorithm achieves the highest delivery rate while maintaining the lowest latency. Although the results indicate that the QLCR routing algorithm may not have the lowest average hop count, it remains within an acceptable range when compared to other algorithms. These findings highlight the effectiveness of the QLCR algorithm in optimizing message delivery performance in DTNs.

In future research, more in-depth investigations will be conducted on factors that influence annotation results, such as features and rules, with the aim of identifying features with greater practical value and resolution rules with stronger resolving power. This will further enhance the overall performance of the proposed method.

## Figures and Tables

**Figure 1 sensors-23-06131-f001:**
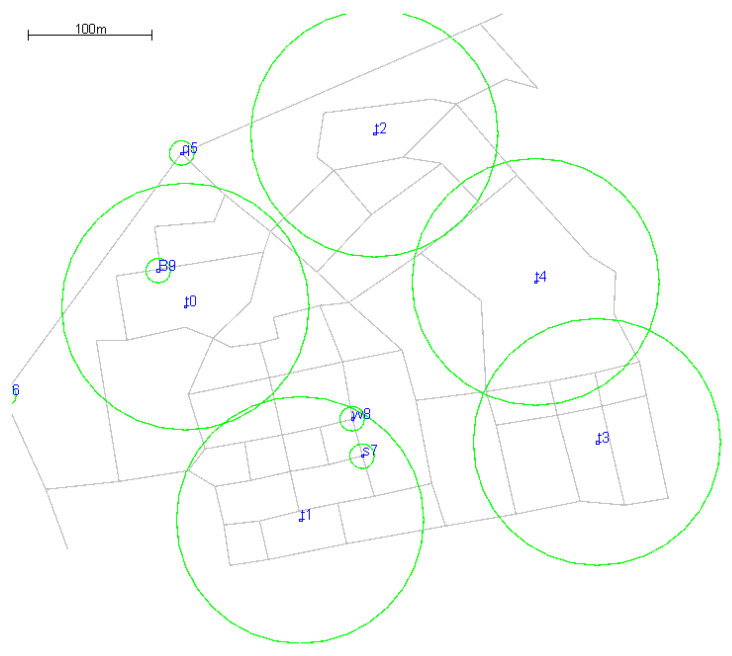
Node motion map and division of interest area.

**Figure 2 sensors-23-06131-f002:**
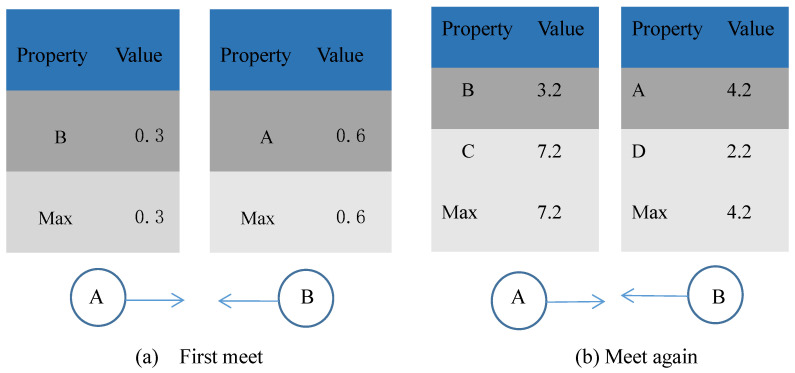
Update process diagram of Q table.

**Figure 3 sensors-23-06131-f003:**
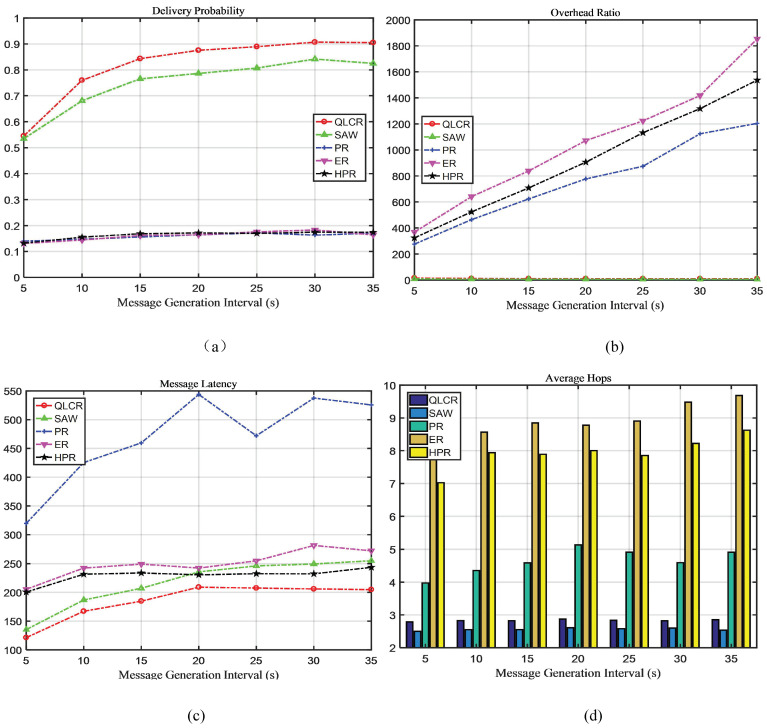
The impact of message generation interval on network performance.

**Figure 4 sensors-23-06131-f004:**
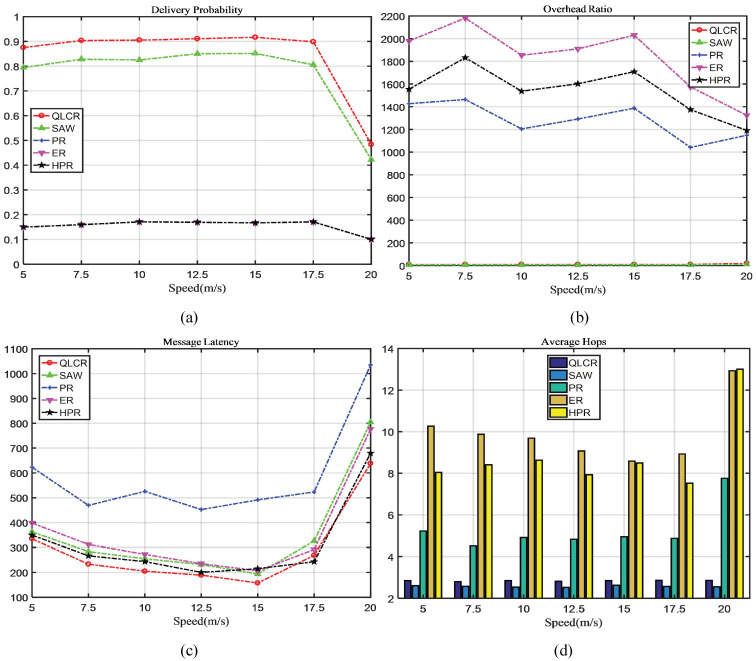
The impact of node speed on network performance.

**Figure 5 sensors-23-06131-f005:**
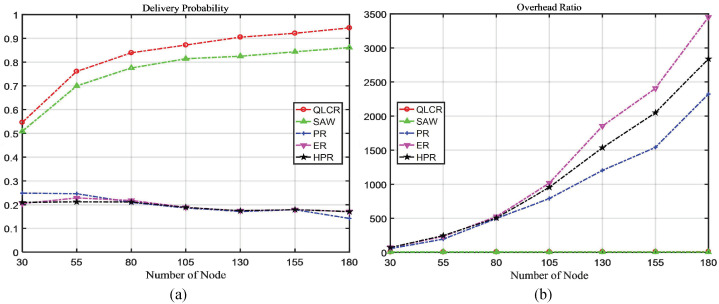
The impact of node count on network performance.

**Figure 6 sensors-23-06131-f006:**
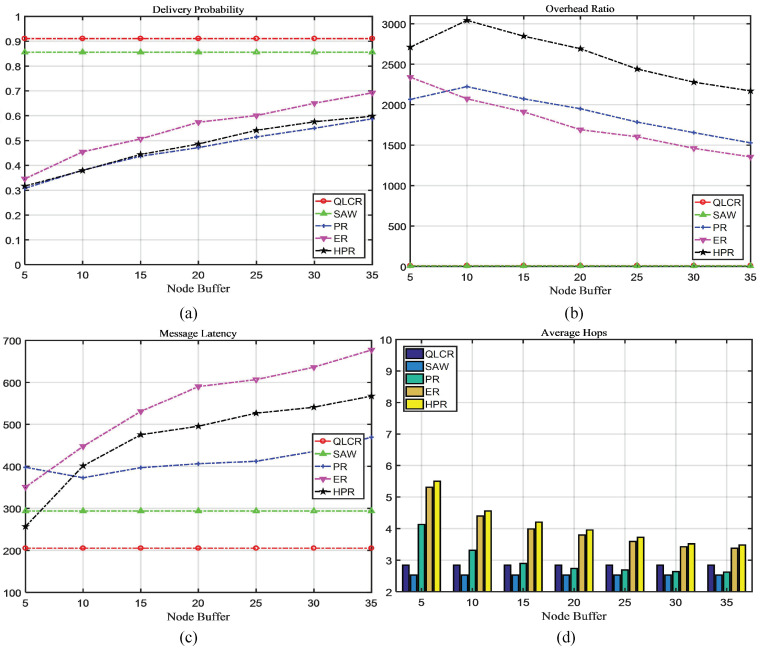
The impact of node buffer on network performance.

**Table 1 sensors-23-06131-t001:** A brief overview of algorithms involved.

Algorithm Involved	Main Feature
ER	Utilizes a flooding method for information transmission.
SAW	Low transmission delay, close to optimal.
PR	Introduces a predictive delivery probability function and uses the probability value as the selection condition for relay nodes.
HPR	HPR algorithm does not select relay nodes, but serves as a comparison to the algorithm proposed in this article.
QLCR	Refers to the concept of community division to determine appropriate relay nodes.

**Table 2 sensors-23-06131-t002:** Simulation configurations.

Parameters	Values
Movement model	Map-based
Buffer size	5–10 M
Wait time	0–120 s
Maximum speed	20 ms^−1^
Number of nodes	30–180
Event generators	5–35 s
Transmit range	10 m
Transmit speed	10 Mbps
Group number	10
Message TTL	300 s

## Data Availability

Not applicable.

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
