# Peer review of "Multi-Copy Relay Node Selection Strategy Based on Reinforcement Learning"

_sensors, 2023, doi:10.3390/s23136131_

Round 1
Reviewer 1 Report
The authors suggest using Q-lambda reinforcement learning in the multi-copy relay node selection algorithm demonstrates an innovative approach to improving delivery rates and reducing message delays in DTNs.
The paper is interesting but needs to be in scientific, rigorous form to be published.
The text needs to be reviewed considering spaces inter-words, including the abstract. For example: "Delay tolerant networks(DTNs)", "delays.To".
The abstract should be reduced.
The related work section needs to present a summary of the proposals. A table could be used.
The Introduction section does not clarify why a new protocol must be proposed. It is necessary to show strong motivation.
Some acronyms were not defined before use. For example: "HPR algorithm"
Algorithm 2 shows incorrect formatting.
Section 3.2 can be summarized in equations instead of text.
Line 235: "(QLCR, PR, EP, HPR, and SAW" - EP or ER? Epidemic Routing.
Is the code available? What is The ONE simulator version?
The consumption of Q-learning techniques could impede its adoption. It is necessary to discuss it.
It is necessary to discuss the adoption of QLCR instead of SAW once the last one is less overhead and uses fewer hops.
Line 270 needs to be clarified: "5. Conclusions - This section is not mandatory, but can be added to the manuscript if the discussion is unusually long or complex"
The results need to be discussed more.
Some articles need to be cited. For example:
[1] de Oliveira, E.C., Silva, E.F., Passos, D., Naves, J., Muchaluat-Saade, D.C., Moraes, I.M. and Albuquerque, C., 2016. Context-aware routing in delay and disruption tolerant networks. International Journal of Wireless Information Networks, 23, pp.231-245.
[2] Visca, Jorge, and Javier Baliosian. "rl4dtn: Q-Learning for Opportunistic Networks." Future Internet 14, no. 12 (2022): 348.
I suggest a review by a native speaker.
Author Response
Comment 1.1:The authors suggest using Q-lambda reinforcement learning in the multi-copy relay node selection algorithm demonstrates an innovative approach to improving delivery rates and reducing message delays in DTNs.
The paper is interesting but needs to be in scientific, rigorous form to be published.
Response 1.1:So many thanks for your affirmation and approval to the article. We have addressed all the concerns raised by the reviewers. Detailed responses to those comments are given below.
Comment 1.2:The text needs to be reviewed considering spaces inter-words, including the abstract. For example: "Delay tolerant networks(DTNs)", "delays.To".
Response 1.2:Thank you for your valuable feedback and suggestions. All these issues have been addressed. We have checked and corrected the entire article.
Comment 1.3: The abstract should be reduced.
Response 1.3: Thank you for your valuable feedback and suggestions. We have removed unnecessary sentences from the abstract section.
Comment 1.4: The related work section needs to present a summary of the proposals. A table could be used.
Response 1.4: Thank you for your valuable feedback and suggestions. In the related work section, we have established a table to display the algorithms involved and provided explanations. (Section 2, page 4)
Comment 1.5: The Introduction section does not clarify why a new protocol must be proposed. It is necessary to show strong motivation.
Response 1.5: Thank you for your valuable feedback and suggestions. To highlight the necessity of our proposed algorithm, we have added an explanation in the introduction section. (section 1, page 2, paragraph 3)
Comment 1.6:Some acronyms were not defined before use. For example: "HPR algorithm"
Response 1.6:We have addressed this issue. For example we have explained where the HPR algorithm first appeared. (Section 2, page 3, paragraph 1)
Comment 1.7:Algorithm 2 shows incorrect formatting.
Response 1.7: Thank you for your valuable feedback and suggestions. Now we have revised the format of Algorithm 2. (Section 3, page 8)
Comment 1.8:Section 3.2 can be summarized in equations instead of text.
Response 1.8: Now we have replaced the text with Pseudocode according to your suggestions, and we have also defined the relevant symbols clearly. (Section 3.2, page 6)
Comment 1.9:Line 235: "(QLCR, PR, EP, HPR, and SAW" - EP or ER? Epidemic Routing.
Response 1.9: Thank you for your valuable feedback and suggestions. It should be ER.We have corrected this error.
Comment 1.10:The consumption of Q-learning techniques could impede its adoption. It is necessary to discuss it.
Response1.10:Thank you for your valuable feedback and suggestions. We have analyzed the limitations of the Q-learning reinforcement learning method in the corresponding sections a ccording to the suggested modifications.(Section 2, page 3, paragraph 5)
Comment 1.11:It is necessary to discuss the adoption of QLCR instead of SAW once the last one is less overhead and uses fewer hops.
Response1.11:We have discussed and explained, based on your valuable feedback, that des pite our algorithm performing worse than the SAW algorithm in terms of average hop count, there are still reasons why our algorithm can be adopted. (Section 5, page 13,paragraph 2)
Comment 1.12:Line 270 needs to be clarified: "5. Conclusions - This section is not mandatory, but can be added to the manuscript if the discussion is unusually long or complex"
Response1.12: Thank you for pointing out that this sentence was a leftover from the templa te and was not properly removed. We sincerely apologize for this oversight. We have now deleted this sentence from the document.
Comment 1.13:The results need to be discussed more.
Response1.13:Thank you for your valuable feedback. Firstly, we have added a new discussion section, and secondly, we have expanded the conclusion section. (Section 5 and Section 6)
Comment 1.14:Some articles need to be cited. For example:
[1] de Oliveira, E.C., Silva, E.F., Passos, D., Naves, J., Muchaluat-Saade, D.C., Moraes, I.M. an
d Albuquerque, C., 2016. Context-aware routing in delay and disruption tolerant networks. I
nternational Journal of Wireless Information Networks, 23, pp.231-245.
[2]Visca, Jorge, and Javier Baliosian. "rl4dtn: Q-Learning for Opportunistic
Networks." Future Internet 14, no. 12 (2022): 348.
Response1.14:Thank you for your valuable feedback. We have studied the article you mentioned, and we have benefited greatly from it. We have now cited these two articles, numbered as [32] and [33].

Reviewer 2 Report
The authors in this work present an idea about the transmitten a message DTN network.
The solution is based how to identify the good relay from the community. So, they propose to use the degree and the interest of relay like SON. The mecanism is motivated based Q-learning technique to estimate the average reward. The model is formulated and the simulation resutls are given with comments.
There are some reamarks:
1- In DTN network, the delay is normal for some applications, but the authors can add a constraint on the TTL like the valide time to be respected by the relay .
2- The buffer is key factor also for the relay to manage all stored messages and they influenced by the energy consummed and also by the message selected, in this case, how the relaly select one message to be transmitted at each conact time.
3. The paper need more improvement in presentation form, like in absract and and --> and
4. The authors can add some references for readers
- Y. Larabi, K. Ibrahimi , J. Ben-Othman and E. Amhoud, A Complete Transmitted Message in DTNs with a Stable Coalition in Dynamic Structures. To appear in the Proceedings of The International Wireless Communications And Mobile Computing Conference (IWCMC), June 19 - 23, 2023 Marrakesh, Morocco - S. Burleigh et al., "Delay-tolerant networking: an approach to interplanetary Internet," in IEEE Communications Magazine, vol. 41, no. 6, pp. 128-136, June 2003, doi: 10.1109/MCOM.2003.1204759.- S. Mekouar, K. Ibrahimi and E. -H. Bouyakhf, "Inferring trust relationships in the social network: Evidence theory approach," 2014 International Wireless Communications and Mobile Computing Conference (IWCMC), Nicosia, Cyprus, 2014, pp. 470-475, doi: 10.1109/IWCMC.2014.6906402.
The paper need a minor improvement in Enlish
Author Response
Multi-copy relay node selection strategy based on reinforcement learning
Yang Gao ,Changjiang Shi,Fuquan Zhang*
We would like to thank the editor and the anonymous reviewers for their invaluable comments, suggestions and feedback to improve this manuscript. We have addressed all the concerns raised by the reviewers. Detailed responses to those comments are given below. Note that the page numbers correspond to the new version of the manuscript. A diff file highlighting the updates(the deleted text appears in red and added text appears in blue.) relation to the original submission is also submitted.
Response to comments from reviewer 2
- Comment 2.1:The solution is based how to identify the good relay from the community. So, they propose to use the degree and the interest of relay like SON. The mecanism is motivated based Q-learning technique to estimate the average reward. The model is formulated and the simulation resutls are given with comments.
- Response 2.1:So many thanks for your affirmation and approval to the article. We have addressed all the concerns raised by the reviewers. Detailed responses to those comments are given below.
- Comment 2.2:In DTN network, the delay is normal for some applications, but the authors can add a constraint on the TTL like the valide time to be respected by the relay .
- Response 2.2:Thank you for your valuable feedback.In the actual operation of the algorithm, the message has a lifetime. We have added the values of relevant parameters in Table 2. (Section 4, page 8, table 2)
- Comment 2.3:The buffer is key factor also for the relay to manage all stored messages and they influenced by the energy consummed and also by the message selected, in this case, how the relaly select one message to be transmitted at each conact time.
- Response 2.3: Thank you for your valuable feedback.This algorithm adopts a first in, first out message transmission mechanism within a link time. We have added explanations in relevant positions in the text. (Section 3, page 8, paragraph 1)
- Comment 2.3:The paper need more improvement in presentation form, like in absract and and --> and
- Response 2.3: Thank you for your valuable feedback.Based on your valuable feedback, we have reorganized the structure and layout of the entire text. Thank you very much for checking so carefully.
- Comment 2.4:The authors can add some references for readers.
[1]Y. Larabi, K. Ibrahimi , J. Ben-Othman and E. Amhoud, A Complete Transmitted Message in DTNs with a Stable Coalition in Dynamic Structures. To appear in the Proceedings of The International Wireless Communications And Mobile Computing Conference (IWCMC), June 19 - 23, 2023 Marrakesh, Morocco
[2] S. Burleigh et al., "Delay-tolerant networking: an approach to interplanetary Internet," in IEEE Communications Magazine, vol. 41, no. 6, pp. 128-136, June 2003, doi: 10.1109/MCOM.2003.1204759.
[3]S. Mekouar, K. Ibrahimi and E. -H. Bouyakhf, "Inferring trust relationships in the social network: Evidence theory approach," 2014 International Wireless Communications and Mobile Computing Conference (IWCMC), Nicosia, Cyprus, 2014, pp. 470-475, doi: 10.1109/IWCMC.2014.6906402.
- Response 2.4: Thank you for your valuable feedback.I'm really sorry, I couldn't find the first article you mentioned. Today is the time for me to revise my paper and submit it for the second time. Today is June 19th, 2023, and the meeting for the article you are referring to is only held today.The order numbers of the other two articles are 14 and 19.

Reviewer 3 Report
What is N in N2NC_i = {1, 2, . . . , N}? The notation N should be defined first.
In Eq.(2), what is s_i and what is a_i? The set of states and the set of actions should be defined first.
dv => d_v?
If A and B are good friends , B will usually be closer than A and B. So for this point B, it’s more similar to A than A is to B. => need some more revision to make it clearly understood.
Eq(1) is symmetric. Do you use asymmetric similarity?
In Eq(2), what is S and What is A?
In Algorithm 1, what are a_d, s_d, a_i, and s_i?
if ( thenThe update Q-value is the Maximum value in the Q-table) Message transfer; => if ( The update Q-value is the Maximum value in the Q-table) then Message transfer;
In Fig.2(b), what is the MAX 8.2 rather than 7.2?
In Fig.3(b) and Fig.4(b), what is the definition of overhead ratio?
In Fig.3(c) and Fig.4(c), what is the time unit of y-axis?
fig. => Fig. There are many places with this typo.
This section is not mandatory, but can be added to the manuscript if the discussion is unusually long or complex. => This sentence should be removed.
There are many places where there is no space when a space is needed. For example,
- nodes.Based
- division(QLCR).In community
- most degree,it is
Divide the population into different types (students, residents, teachers, others) and limit the movement range of different types of people. => The sentence does not have a subject.
Divide ideas by community, the nodes within a community are much denser than those between communities, which implies that each node and the majority of its neighbors should belong to the same community. => The sentence does not have a subject.
Author Response
Multi-copy relay node selection strategy based on reinforcement learning
Yang Gao ,Changjiang Shi,Fuquan Zhang*
We would like to thank the editor and the anonymous reviewers for their invaluable comments, suggestions and feedback to improve this manuscript. We have addressed all the concerns raised by the reviewers. Detailed responses to those comments are given below. Note that the page numbers correspond to the new version of the manuscript. A diff file highlighting the updates(the deleted text appears in red and added text appears in blue.) relation to the original submission is also submitted.
Response to comments from reviewer 3
- Comment 3.1:What is N in N2NC_i = {1, 2, . . . , N}? The notation N should be defined first.
- Response 3.1:I deeply apologize for the confusion. It should not be N, but rather m, and m has already been defined in advance.We have made the necessary corrections in the respective section of the article.
- Comment 3.2:In Eq.(2), what is s_i and what is a_i? The set of states and the set of actions should be defined first.
- Response 3.2:Thank you for your valuable feedback. We have made the necessary modifications to the equation and clearly defined all the symbols involved in the article. (Section 3, page 6, Eq.(3))
- Comment 3.3:dv => d_v?
- Response 3.3:Thank you for your valuable feedback and suggestions. It should indeed be d_v, and we have made the necessary modification in the corresponding section of the article.
- Comment 3.4:If A and B are good friends , B will usually be closer than A and B. So for this point B, it’s more similar to A than A is to B. => need some more revision to make it clearly understood.
- Response 3.4: Thank you for your valuable feedback. To clarify this issue, we have made further revisions to the relevant principles and added an equation in the article. (Section 3, page 5)
- Comment 3.5:Eq(1) is symmetric. Do you use asymmetric similarity?
- Response3.5Thank you for raising this valuable question. Although the equation is symmetric, we can modify it into two separate equations. We have already added the relevant equations to the article. Furthermore, we have provided a description of the relevant concepts. (Section 3, page 5, formula 2)
- Comment 3.6:In Eq(2), what is S and What is A?
- Response 3.6:In order to facilitate understanding and make it more intuitive, we have revised this formula and provided specific explanations for the symbols in the formula. ( Section 3, page 5, formula 3)
- Comment 3.7:In Algorithm 1, what are a_d, s_d, a_i, and s_i?
- Response 3.7:Thank you for your valuable feedback. We have provided explanations for the symbols a_d, s_d, a_i, and s_i in Algorithm 1. (Section 3, page 7, Algorithm 1)
- Comment 3.8:if ( thenThe update Q-value is the Maximum value in the Q-table) Message transfer; => if ( The update Q-value is the Maximum value in the Q-table) then Message transfer;
- Response 3.8:Thank you for your thorough review. I can see the gaps in my understanding from your feedback.The correct result is : if (The update Q-value is the Maximum value in the Q-table) then Message transfer; We have made revisions to Algorithm 1 to address this issue.
- Comment 3.9:In Fig.2(b), what is the MAX 8.2 rather than 7.2?
- Response 3.9:Thank you for your meticulous examination. The correct result should indeed be 7.2. We have made the necessary corrections at the respective location in the article.
- Comment 3.10:In Fig.3(b) and Fig.4(b), what is the definition of overhead ratio?
- Response 3.10:Thank you for your valuable feedback. This issue has now been addressed. According to your advice we have defined overhead (Section 4, page 8-9)
- Comment 3.11:In Fig.3(c) and Fig.4(c), what is the time unit of y-axis?
- Response 3.11:Thank you for your valuable feedback. The unit of message delay is seconds. We have added the definition and unit of message delay in the document.(Section 4, page 9, paragraph 2)
- Comment 3.12: => Fig. There are many places with this typo.
- Response 3.12:Thank you for pointing out the issue. We have identified and rectified the problem by changing "f" to "F".
- Comment 3.13:This section is not mandatory, but can be added to the manuscript if the discussion is unusually long or complex. => This sentence should be removed.
- Response 3.13:Thank you for pointing out that this sentence was a leftover from the template and was not properly removed. We sincerely apologize for this oversight. We have now deleted this sentence from the document.

Reviewer 4 Report
The authors have done a novel work my proposing a multi-copy relay node selection router algorithm based on Q-lambda reinforcement learning with reference to the idea of community division (proposes a DTN routing method based on Q-Lambda reinforcement learning)
They went further to analyse their work and compared with existing literature. This attempt give validity of their work and I am impressed with it
However, my comments is for the authors to bullet point their main contributions to the body of knowledge which the author already have. Which should come after after related works.
Also, several work have dwell on Machine learning irrespective of where they were applied . The truth is that the idea remain the same. Looking at these would make the author introduction and related work more broader and robust. I advise the author to kindly look at some. For example
Integrating Artificial Intelligence (A.I), Internet of Things (IoT) and 5G for Next-Generation Smart grid: A Survey of Trends and Prospect” IEEE Access, Vol 10, pp. 4794-4831, 2022.
Overall, this authors have done extremely well hence I strongly recommemded this work for publication once the corrections are adhered to.
The authors have done a novel work my proposing a multi-copy relay node selection router algorithm based on Q-lambda reinforcement learning with reference to the idea of community division (proposes a DTN routing method based on Q-Lambda reinforcement learning)
They went further to analyse their work and compared with existing literature. This attempt give validity of their work and I am impressed with it
However, my comments is for the authors to bullet point their main contributions to the body of knowledge which the author already have. Which should come after after related works.
Also, several work have dwell on Machine learning irrespective of where they were applied . The truth is that the idea remain the same. Looking at these would make the author introduction and related work more broader and robust. I advise the author to kindly look at some. For example
Integrating Artificial Intelligence (A.I), Internet of Things (IoT) and 5G for Next-Generation Smart grid: A Survey of Trends and Prospect” IEEE Access, Vol 10, pp. 4794-4831, 2022.
Overall, this authors have done extremely well hence I strongly recommemded this work for publication once the corrections are adhered to.
Author Response
Multi-copy relay node selection strategy based on reinforcement learning
Yang Gao ,Changjiang Shi,Fuquan Zhang*
We would like to thank the editor and the anonymous reviewers for their invaluable comments, suggestions and feedback to improve this manuscript. We have addressed all the concerns raised by the reviewers. Detailed responses to those comments are given below. Note that the page numbers correspond to the new version of the manuscript. A diff file highlighting the updates(the deleted text appears in red and added text appears in blue.) relation to the original submission is also submitted.
Response to comments from reviewer 4
- Comment1: The authors have done a novel work my proposing a multi-copy relay node selection router algorithm based on Q-lambda reinforcement learning with reference to the idea of community division (proposes a DTN routing method based on Q-Lambda reinforcement learning)
They went further to analyse their work and compared with existing literature. This attempt give validity of their work and I am impressed with it
- Response1:So many thanks for your affirmation and approval to the article. We have addressed all the concerns raised by the reviewers. Detailed responses to those comments are given below.
- Comment4.2:However, my comments is for the authors to bullet point their main contributions to the body of knowledge which the author already have. Which should come after after related works.
- Response 4.2:Thank you for pointing out the issue. We have moved the contributions to the section "Related Work" and added bullet points for each item.
- Comment4.3:Also, several work have dwell on Machine learning irrespective of where they were applied . The truth is that the idea remain the same. Looking at these would make the author introduction and related work more broader and robust. I advise the author to kindly look at some. For example
Integrating Artificial Intelligence (A.I), Internet of Things (IoT) and 5G for Next-Generation Smart grid: A Survey of Trends and Prospect” IEEE Access, Vol 10, pp. 4794-4831, 2022.
- Response 4.3:This issue has now been addressed. We have included a table after the related works section, showcasing the algorithms involved. Additionally, we have cited the article you mentioned ,numbered as [34].

Round 2
Reviewer 1 Report
The authors improved the paper following the suggestions.
The paper can be accepted in its present form.